# The Influence of Shelter Type and Coverage on Crayfish (*Procambarus clarkii*) Predation by Catfish (*Silurus asotus*): A Controlled Environment Study

**DOI:** 10.3390/ani14081147

**Published:** 2024-04-10

**Authors:** Mingguang Zhao, Guangpeng Feng, Haihua Wang, Chenchen Shen, Yilong Fu, Yanping Zhang, Haixin Zhang, Yuan Yao, Jianhua Chen, Weikang Xu

**Affiliations:** 1East China Sea Fisheries Research Institute, Chinese Academy of Fishery Sciences, Shanghai 200090, China; 13320732837@163.com (M.Z.); 18936152362@163.com (C.S.); 2Jiangxi Fisheries Research Institute, Nanchang 330039, China; jxswhh@163.com (H.W.); fuyilong@126.com (Y.F.); zhangyanpingxie@163.com (Y.Z.); zhang73860@126.com (H.Z.); 15907099328@163.com (Y.Y.); 3School of Marine Science and Fisheries, Jiangsu Ocean University, Lianyungang 222005, China; chenjianhuazsu@163.com (J.C.); 18551002772@163.com (W.X.)

**Keywords:** shelters, *Procambarus clarkii*, *Silurus asotus*, predation, covert behavior

## Abstract

**Simple Summary:**

*Procambarus clarkii* is typically viewed as a predator rather than prey, due to its strong anti-predatory capabilities. However, during its juvenile stage, it remains vulnerable to predators. Nonetheless, complex outdoor habitats can offer some protection. By examining the effects of three different types of shelters with various coverages and combinations on the predation rhythm, behavior, and efficiency of *Silurus asotus* on juvenile *Procambarus clarkii*, we found that the presence and complexity of shelters in the environment have a limited impact on the predatory activities of *Silurus asotus*. Among the shelter types, PVC pipes offered the best protection, followed by stone, with water grass providing the least protection. Furthermore, the protective effect of shelters increased with coverage rate. Therefore, we suggest that the type and abundance of shelters significantly influence the predation rhythm and behavior of *Silurus asotus* on *Procambarus clarkii* in outdoor environments.

**Abstract:**

*Procambarus clarkii* is adept at using natural shelters and caves to evade attacks from predators. However, the concealment abilities and mechanisms of *P. clarkii* for different types of shelters under predation pressure have not yet been reported. In this study, laboratory experiments were carried out to determine the effects of different coverages (25%, 50%, and 75%) and different combinations (I–VII) of three types of shelters (PVC pipes, water grass, and stone) on the predation rhythm, behavior, and abilities of *Silurus asotus* on *P. clarkii.* The results indicated that the predation of *S. asotus* on *P. clarkii* exhibited significant rhythmicity under shelter conditions, excluding PVC pipes, 75% stone, and combination VI. Among the three types of shelters, PVC pipes provided the strongest concealment, followed by stone and water grass. With the increase in shelter coverage, the anti-predation ability of *P. clarkii* continued to increase, and the optimal shade rate for water grass was 50%. In the different shelter combinations, the environmental complexity had little effect on the predation activity of *S. asotus* on *P. clarkii*. These findings demonstrated that the type and abundance of shelters in the wild environment can affect the predation rhythm and activities of *S. asotus* on *P. clarkii.*

## 1. Introduction

*Procambarus clarkii* is a typical cave-dwelling animal with social behavior [1], known for its aggressiveness, burrowing habits, and robust anti-predation ability [2]. In the late 1920s, *P. clarkii* was introduced to China from Japan, and it has since established a wide distribution in the Yangtze River Basin, occupying an ecological niche in Poyang Lake [3]. From 2009 to 2013,the production of *P. clarkii* in Poyang Lake represented over 50% of the total shrimp fishing production, averaging 2.5 × 10^4^ tons [4]. However, following the ban on fishing in the Yangtze River in 2021, *P. clarkii* in Poyang Lake exhibited a trend towards smaller individuals and a substantial decline in resources. For example, the results of a 2021~2022 survey indicated that the presence of *P. clarkii* was detected at only 5 of the 14 sampling sites, and the number of *P. clarkii* caught was significantly lower than before the closure [5]. In nature, the predation activities of numerous predators, including waterfowl, water snakes, and some fierce carnivorous fish (*Anguilla anguilla*, *Silurus asotus*, *Perca fluviatilis*, etc.) may contribute to the decline in *P. clarkii* resources [6,7,8]. The interactions between prey size and predator species can influence the feeding preferences of the predator [9]. For example, predators exhibit a clear preference for the size of *Penaeus orientalis* [10] and *Cherax quadricarinatus* [11]. For *P. clarkii*, the small larvae may face a higher risk of predation.

The natural shelters in the environment offer protection for prey from the threat of predation. For *P. clarkii*, various effective shelter types include substrates, caves, large plants [12], and even shadows [13]. The specifications of shelters are a crucial factor affecting their protective capacity, with larger shelters typically providing better protection for *P. clarkii* and influencing their efficiency in responding to predators [14]. The effectiveness of shelters is proportional to their density and complexity [15,16]. Shelters can reduce the frequency and duration of interspecies competition and the risk of interspecies predation, significantly improving the survival rate of *P. clarkii* [1,13]. Shelters are more effective in reducing the aggression of *Cambarus* than food [17]. *P. clarkii* exhibits an obvious preference for different types of shelters. Regarding shelter type, *P. clarkii* preferred narrow, dark, and breathable shelters, showing a clear selective preference for cave-type shelters [18]. In terms of shelter materials, *P. clarkii* preferred bamboo-type shelters [19].

However, shelters cannot consistently yield a positive effect for *P. clarkii*, whether in the natural environment or aquaculture process. In the environments with shelters, the different developmental stages of *P. clarkii* will compete for shelters, leading to the larvae of *P. clarkii* being confined to areas with sparse shelters, increasing the risk of predation [20]. Simultaneously, in order to reduce the probability of interspecies fighting and predation, *P. clarkii* feed only to satisfy their own minimum energy expenditure, resulting in diminished growth rates [21]. Additionally, some shelters like straw, mud, and dead leaves may decrease the hatching rate of fertilized oosperm [22]. Currently, numerous studies have focused on the effects of shelters on the growth of *P. clarkii*, but there is limited research on the impact of shelters on its other life activities. Poyang Lake is a typical representative of composite river and lake ecology, with a complex habitat and numerous predators. Whether shelters will affect the anti-predation activity of *P. clarkii* and the mechanism of action remain unknown.

In this study, *P. clarkii* were used as the prey and *Silurus asotus* were selected as the predator. Three types of shelters (PVC pipes, water grass, and stone) were used to simulate diverse habitat conditions in Poyang Lake. Moreover, the different coverages and combinations of three shelters were set up to analyze the feeding rhythm, anti-predation behavior, and activity of *P. clarkii*. These results will reveal the influence mechanism of different habitats on the anti-predation activity of *P. clarkii* and provide a scientific reference for the resource assessment and sustainable development of *P. clarkii* in Poyang Lake.

## 2. Materials and Methods

### 2.1. Source and Maintenance of Experimental Animals

The morphological indices of *P. clarkii* and *S. asotus* were shown in Table 1 (with mean ± standard error), which were taken from the aquaculture base of the Jiangxi Fisheries Research Institute (Nanchang, Jiangxi). Before the test, *P. clarkii* and *S. asotus* were separately maintained in 84 indoor aquaria (length 1 m × width 0.6 m × height 0.55 m, water depth 0.15 m, each equipped with a recirculating system) for two weeks. During the acclimation, water subjected to continuous aeration for more than 48 h was provided, and an appropriate amount of sediment was added to the water to simulate the water transparency of Poyang Lake during the dry water period. The water temperature was maintained at 23 ± 0.5 °C, with a photoperiod of 12 h dark:12 h light, pH of 7.5 ± 0.2, and dissolved oxygen ≥ 6 mg/L. Commercial crayfish pellets (32% crude protein, 4% crude lipid, 8% crude fiber, 3% calcium, 1% phosphorus, 1.5% lysine, 15% ash and 14% moisture) were used to feed *P. clarkii* every three days, and *P. clarkii* of the above size were employed to feed the *S. asotus* to saturation once every other day.

### 2.2. Experimental Design and Procedure

Three levels of shelter coverage (25%, 50%, 75%) and seven different combinations of three types of shelters (PVC pipes: φ 0.028 m, length 0.1 m; water grass: length 0.1 m; stone: length 0.08 m, width 0.06 m) with 50% coverage were set up (Table 2), including 14 groups with three replicates each, and the control group without shelters. The experiment lasted for 24 h. All experimental conditions were consistent with the acclimation and the photoperiod consisted of 12 h light and 12 h dark. A transparent perforated acrylic plate separating areas A and B of the blue test device (Figure 1) was used, and two cameras were mounted directly above and at the bottom. A small flushing device (length 0.4 m × width 0.75 m × height 0.8 m) was installed in front to simulate the water flow rate.

The 16 *P. clarkii* and one *S. asotus* subjected to 48 h starvation were randomly placed in areas A–B of the test device for two hours before the experiment. After two hours, the plate was withdrawn and the video recording system was activated to document the experiment. The remaining number of *P. clarkii* in the test device was recorded at 12 h intervals. After 24 h, the number of prey of *P. clarkii* was counted. The number of attacks and predation time of *S. asotus* were recorded and analyzed for the predation behavior of *S. asotus* and the anti-predation behavior of *P. clarkii* in the recorded video.

### 2.3. Data Statistical Analysis

The behavior of *S. asotus* when approaching, attacking, and handling *P. clarkii* was considered as a complete predation event. The number of prey (NOP), relative predation rate (RPR) of *P. clarkii*, and the success rate of foraging (SROF) of *S. asotus* were calculated as follows:$$\mathrm{NOP}=N_{0}-Ni$$
SROF=NOP/NOA×100%
$$\mathrm{RPR}=\mathrm{NOP}/\left(N_{0}\times D\right)\times 100\%$$
where, N_0_: Initial number of *P. clarkii*, N_i_: Remaining number of *P. clarkii*, D: Number of predators, NOA: Number of attacks.

The results were expressed as mean ± standard error (SE). IBM SPSS Statistics 27.0 was used to conduct one-way ANOVA and multiple comparisons to analyze differences between groups, *p* < 0.05 indicates a significant difference. Results were plotted with Graphpad Prism 8.

## 3. Results

### 3.1. Predation Rhythm

The predation results of daytime and night (12 h:12 h) indicated that the RPR for the other experimental groups of *S. asotus* on *P. clarkii* was significantly higher at night (*p* < 0.05) with the exception of 25% and 75% PVC pipes, 75% stone, and combination VI (Table 3). Similarly, except for 75% PVC pipes, the NOA of *S. asotus* was significantly higher at night (*p* < 0.05) (Table 3).

### 3.2. Predation and Anti-Predation Behavior

Fish aggression was an adaptive behavior exhibited by individual animals of the same or different species to compete for space or food resources, which was an innate and prominent characteristic of fish [23]. Predation strategies and behaviors of *S. asotus* were found to be consistent across species and coverage of shelters during the course of the experiment. Sample images from videos, depicting the predation behavior of *S. asotus* on *P. clarkii*, are shown in figure. 2. Based on the analysis of the video recordings, the predation process of *S. asotus* on *P. clarkii* was divided into three stages, including the search, attack, and prey handling. Stage I was the search stage, in which *S. asotus* persisted in searching for prey along the wall of the test device and most *P. clarkii* remained stationary on the wall or in a corner during the attack latency. Stage II was the attack stage. Upon contact with the *P. clarkii*, the *S. asotus* immediately accelerated and attacked the body of *P. clarkii* (Figure 2A,B). Simultaneously, the *P. clarkii* rapidly raised their *pedes chela* and swung their tails to escape along the wall or toward the opposite direction. It remained stationary on another safe corner or wall if successful, and *S. asotus* would continue to swim along the tank wall in search of the next prey. Stage III was the prey handling stage. After capturing the prey, *S. sotus* gradually swallowed it for chewing upon capturing the prey, and consistently pushed the prey against the wall and shook its body rhythmically until it completely devoured its prey during the swallowing process (Figure 2C,D). After finishing its meal, *S. asotus* left along the wall of device.

### 3.3. Effects of Different Coverages of Shelter on the Anti-Predation of P. clarkii

Under the effect of three types of shelters with varying coverage rates, except for 25% stone, the RPR of *P. clarkii* in the remaining experimental groups was significantly lower than that in the control group (*p* < 0.05) (Figure 3). The NOA of *S. asotus* on *P. clarkii* in the remaining experimental groups was significantly lower than that in the control group with the exception of 25% PVC pipes and 75% water grass and 25% stone (*p* < 0.05) (Figure 4). And, apart from the 25% PVC pipes, 50% stone, and 75% water grass, the SROF of *S. asotus* on *P. clarkii* in the remaining experimental groups was significantly lower than that in the control group (*p* < 0.05) (Figure 5)

In the presence of water grass sheltering, the RPR of *P. clarkii* with 50% coverage was significantly lower than that with 25% and 75% coverage (*p* = 0.0008; *p* = 0.0021) (Figure 3). The NOA of *S*. *asotus* on *P. clarkii* showed no significant difference under three coverage rates (*p* > 0.05) (Figure 4). The SROF of *S*. *asotus* on *P. clarkii* with 50% coverage was significantly lower than that with 25% coverage (*p* = 0.0143) (Figure 5). In the presence of PVC pipe sheltering, there was no significant difference in the RPR and SROF under three coverage rates (*p* > 0.05) (Figure 3 and Figure 5). The NOA with 75% coverage was significantly lower than that with 25% and 50% coverage (*p* < 0.0001; *p* = 0.0201) (Figure 4). In the presence of stone sheltering, the RPR and NOA of 50% and 75% coverage were significantly lower than that of 25% coverage (*p* < 0.05) (Figure 3 and Figure 4) and the SROF with 75% coverage was significantly lower than that with 25% coverage (*p* = 0.0009) (Figure 5).

The RPR, NOA, and SROF of *S*. *asotus* on *P. clarkii* with PVC pipes were consistently lower than those of water grass and stone at an equal coverage rate. Specifically, at the coverage rate of 25%, the RPR, NOA, and SROF of PVC pipes were significantly lower than those of water grass and stone (*p* < 0.05) (Figure 3, Figure 4 and Figure 5). The NOA of PVC pipes was significantly lower than that of water grass at the coverage rate of 50% (*p* = 0.0031) and there was no significant difference in the SROF of three types of shelters (*p* > 0.05) (Figure 5). At the coverage rate of 75%, the RPR, NOA, and SROF of PVC pipes and stone were significantly lower than those of water grass (*p* < 0.05) (Figure 3, Figure 4 and Figure 5).

As seen above, 25%, 50% and 75%, respectively, represent the coverage of the three kinds of shelters. The control is the control group, the same applies below.

### 3.4. Effects of Different Combinations of Shelter on the Predation of S. asotus on P. clarkii

In the combinations of different shelters with a coverage rate of 50%, the RPR of seven combinations was significantly lower than that of the control group (*p* < 0.05) (Figure 6). Except for combination I, the NOA of *S*. *asotus* on *P. clarkii* in the remaining combinations was significantly lower than that in the control group (*p* < 0.05) (Figure 7). With the exception of combination V, the SROF of *S*. *asotus* on *P. clarkii* in the remaining combinations was significantly lower than that in the control group (*p* < 0.05) (Figure 6).

Within three single shelter combinations (I, II, III), there was no significant difference in the RPR and SROF of three combinations (*p* > 0.05) (Figure 6), The NOA of combination II was significantly lower than that of combination I (*p* = 0.0139) (Figure 7) and there was no significant difference in the RPR NOA and SROF among combinations IV–VI (*p* > 0.05) (Figure 6 and Figure 7). The RPR, NOA, and SROF showed no significant differences with the increasing type of shelters among the four groups (I, IV, V, VII) containing water grass. Similarly, no notable distinctions were observed among the intergroups that included stone (II, IV, VI, VII) and PVC pipes (III, V, VI, VI) (Figure 6 and Figure 7).

## 4. Discussion

### 4.1. Predation Rhythm of S. asotus under Different Shelter Conditions

The photoperiod can directly or indirectly regulate the diurnal cycle of dissolved oxygen and temperature of water, and in turn, affects the feeding activity of fish [24,25]. For example, the feeding frequency and amount of *Cyprinus carpio* and Abalone can be significantly affected by the photoperiod [26,27]. In this study, the total predation activity of *S. asotus* on *P. clarkii* in all shelter conditions except 25%, 75% PVC tube, 75% stone, and combination VI showed a clear predation rhythm, consistent with the negatively correlated feeding intensity of *Channa argus* with illumination [28]. *S. asotus*, which mainly relies on olfaction and mechanical touch for predation, is a typical demersal fish with negative phototropism for light [29]. Nevertheless, the photoperiod can also assist some fishes that rely on vision for predation in better locating their prey. The reason for the differences in daytime and night feeding rhythms among the five groups mentioned above may be related to the type and coverage rates of shelters. The *P. clarkii* has a sufficient number of shelters to avoid the potential attack threat of the predator after addressing the predation threat with a high coverage rate of PVC pipes, reducing interactions with *S*. *asotus*, consequently affecting the predation rhythm of the *S*. *asotus*.

### 4.2. Effects of Different Coverage of Shelter on the Predation of S. asotus on P. clarkii

The change in the habitat environment can change the predator’s ability to search and process prey [30,31] and the structure of the hidden area of prey [32]. In this study, the RPR, NOA, and SROF of *S. asotus* on *P. clarkii* were reduced to varying degrees by the presence of the three shelters. Among them, the inhibition ability of PVC pipes was the strongest, followed by stone and water grass, which may be related to the concealment ability of shelters and the habitat habits of *P. clarkii* [33].

*P. clarkii* cling to water grass, using it to hide and escape from predators when threatened by predators [34]. It was found that both the RPR and SROF of *S*. *asotus* on *P. clarkii* exhibit a trend of initially decreasing and then increasing with the increase in the coverage rate of shelter in this study. This may be attributed to the insufficient coverage rate of water grass, so that *P. clarkii* had no enough shelters for concealment. Some *P. clarkii* can only stay on the device wall or corner and expose to *S*. *asotus*, increasing the predation risk by *S. asotus*. As the coverage rate increased to 50%, there are enough shelters for *P. clarkii* to avoid the pursuit of *S*. *asotus*, resulting in a decrease in the RPR of *P. clarkii* and the NOA of *S*. *asotus*. The increase in the biomass of large plants affects the ability of visual predators to detect prey [35]. As the coverage rate increased to 75%, excess water grass contacted with *S*. *asotus*—which relies on olfactory and mechanical touch for predation—increased the NOA, and provided a good hiding effect for *S*. *asotus* [36], which reduced its exposure opportunities to *P. clarkii*, resulting in a decrease in the anti-predation awareness of *P. clarkii* and an increase in its predation risk. Ostrand et al. proposed that prey species was the key factor affecting the predation efficiency of predators, followed by vegetation density [37]. This may be the reason why the results of this study are inconsistent with Gotceitas’ conclusion that the increase in vegetation density will significantly reduce the predation efficiency of predator [38].

As a typical burrowing animal, *P. clarkii* is timid and prefers solitude [39]. It avoids predation by seeking shelter in the cave after feeling the threat of predation [2]. Similarly, *Palaemonetes pugio* can use coarse wood chips as a cover when facing predatory fish [40]. In this study, the NOA and SROF of *S*. *asotus* on *P. clarkii* decreased with the increasing coverage rates of PVC pipes. This decline was attributed to potential conflicts among *P. clarkii* individuals in the cave with a low coverage rate of PVC pipes, where stronger individuals displaced weaker counterparts from shelters, creating opportunities for *S*. *asotus* to attack *P. clarkii*. As the coverage rate of PVC pipes increased, enough PVC pipes provided sufficient shelter for *P. clarkii*, which reduced its exposure opportunity to *S*. *asotus*. This is consistent with the research conclusions of Gotceitas et al. [38].

The principle of the stone is similar to that of the PVC tube. We found that the *P. clarkii* will lie under the stone and in crevices to avoid the predator’s attack when the type of shelter is stone, This is consistent with the conclusion that *Homarus americanus* can use stone crevices and caves to evade predators [41]. In this study, both the RPR, NOA, and SROF of *S*. *asotus* on *P. clarkii* continued to decline with the increase in the coverage rate of stone. This was due to the fact that *P. clarkii* did not have enough stones to avoid predators when the coverage rate of stone was low, and this phenomenon improved with increasing stone coverage. The existence of stones will also increase the difficulty of *S*. *asotus* predation on *P. clarkii* [42].

### 4.3. Effects of Different Combinations of Shelter on the Predation of S. asotus on P. clarkii

In the wild environment, habitat complexity diminishes the predator’s predation efficiency by inhibiting the predator’s mobility and providing shelter for prey [43,44], The distribution and quality of shelters affects the strength, duration, and outcome of animal interactions [45,46]. However, some studies have indicated that higher environmental complexity is more conducive to ambushing the predator’s predation activities [36]. In this study, as the complexity of the shelter increased, no significant difference was observed between the experimental groups. This implies that the complexity of the shelters had little effect on the predation activity of *S*. *asotus*. The outcome is related to the unique biological characteristics of *S*. *asotus*, which preferred to use the complex environment to ambush prey. Simultaneously, the prey size, anti-predation ability, and defense structure affect the predation activity of *S*. *asotus* on *P. clarkii* [47,48].

### 4.4. Effects of Shelter on the Resources of P. clarkii

Hydrological changes in Poyang Lake significantly impact the distribution and diversity of wetland vegetation [49]. Poyang Lake experiences a dry season from November to March annually [50]. During this period, wetland vegetation is barren, constituting a critical phase for the growth and development of *P. clarkii* larvae [51], while other bait organisms such as *Squaliobarbus curriculus* and *Pseudorasbora parva* are larger [52]. *P. clarkii* larvae encounters heightened predation challenges in response to the predatory attacks of wintering ferocious fish during this period [53]. Its weak anti-predation and burrowing ability renders it incapable of evading attacks from ferocious fish. The vegetation in Poyang Lake from April to October is abundant and aggressive fish continue to pose a threat to *P. clarkii* after emerging from hibernation. Nevertheless, the rich vegetation provides natural shelters for the mature *P. clarkii* with a strong anti-predation ability, allowing it to escape from the fierce fish. At the peak of spawning and juvenile growth of many bait fish at this stage, aggressive fishes preferentially prey on juveniles with low anti-predator capacity when multiple bait organisms coexist in Poyang Lake [54], which reduces the risk of *P. clarkii* predation.

In the wild environment, the seasonal burrowing behavior of *P. clarkii* is primarily concentrated from May to October [55]. Sufficient caves can provide shelter for *P. clarkii*, mitigating the risk of predator attacks during this stage. On the contrary, in November to April, little caves can be used by *P. clarkii* to evade predator attacks, greatly increasing its risk of predation. Furthermore, the burrowing ability of *P. clarkii* positively correlates with its growth [56]. *P. clarkii* larvae exhibit poor burrowing ability. Faced with a predation threat, low-quality caves fail to offer effective shelter, leaving them vulnerable to predator attacks.

### 4.5. Effects of Environmental Conditions on the Predatory Activities of Fishes

The effect of water transparency on the predation activity of fish is directly related to the predation mode of fish [57,58] such as *Lateolabrax japonicus* and other fish relying on vision to locate the prey. Lower water transparency will seriously affect its normal predation activity and reduce its predation efficiency [59]. The water transparency of Poyang Lake changes periodically with seasons, i.e., the water transparency is higher during the abundant water period and lower during the dry water period [49]. In this study, sediment was added to simulate the water transparency of Poyang Lake during the dry water period, which resulted in the experimental results only reflecting the predation mechanism of *S. asotus* on *P. clarkii* during the dry water period.

Both *S. asotus* and *P. clarkii* prefer to live in stone crevices and caves in the field, and engage in life activities in the evening and at night [60,61]. This overlap in habitat and active periods could be a contributing factor to *S. asotus* predating on *P. clarkii*. Meanwhile, many fierce fishes in Poyang Lake migrate to the shores to spawn during the spring, meaning that the *P. clarkii* larvae inhabiting the shallow water and mudflats may face a greater threat of predation [62].

The size of the oral fissure is a crucial factor limiting the feeding preferences of predators [63], the smaller size of *P. clarkii* larvae expands the spectrum of potential predators. According to the Optimal Foraging Theory [64], predators expend significantly more energy attacking adult *P. clarkii* than they gain in return. Conversely, preying on *P. clarkii* larvae optimizes the predator’s energy acquisition [54]. This increased efficiency may contribute to the heightened predation risk faced by *P. clarkii* larvae.

## 5. Conclusions

Generally, this study revealed that different shelter combinations and coverage rates of three types of shelters (PVC pipes, water grass and stone) exerted varying inhibitory effects on the predation activity of *S. asotus* on *P. clarkii*. Among the three types of shelter, PVC pipes exhibit the strongest sheltering ability, followed by stone and water grass. In different shelter combinations, environmental complexity had little impact on the predation activity of *S. asotus* on *P. clarkii*. Due to the size limitations of the experimental device and single species of prey, it is impossible to fully simulate the natural habitat, which may lead to limitations in the experimental results and an incomplete explanation of the predatory mechanisms of ferocious fish on *P. clarkii* in the wild. Additionally, Poyang Lake harbors many ferocious fish species, whose predation may still impact the resources of *P. clarkii*. Future research should focus on the effects of predation by different carnivorous fish on *P. clarkii* to refine the understanding of how ferocious fish predation affects the resources of *P. clarkii*.

## Figures and Tables

**Figure 1 animals-14-01147-f001:**
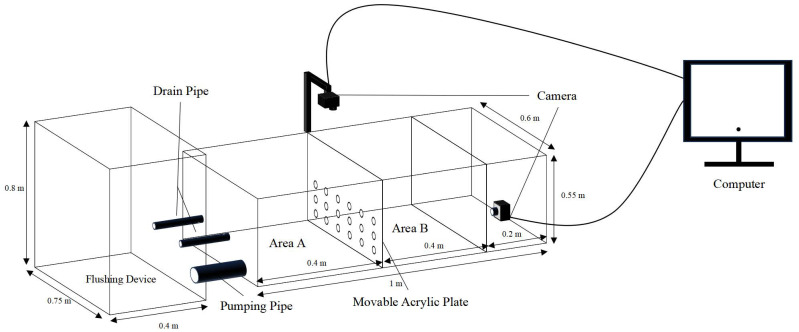
Predator–prey test device.

**Figure 2 animals-14-01147-f002:**
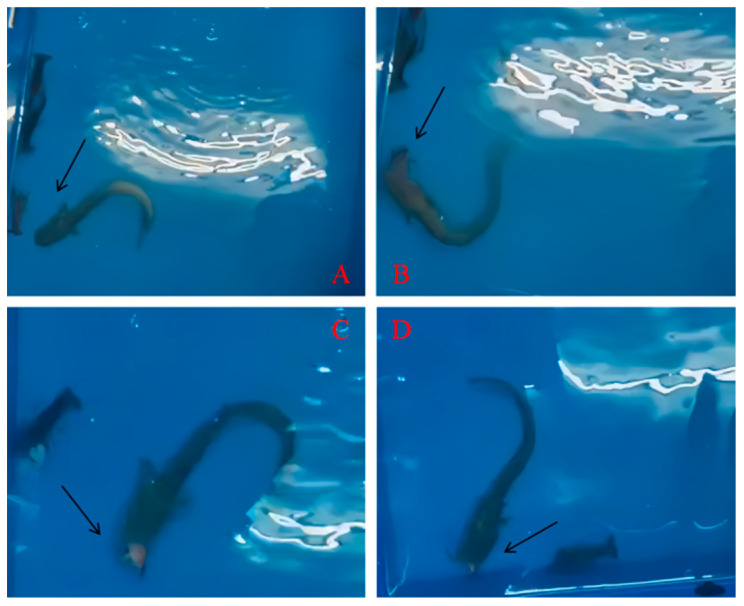
Predatory process of *S*. *asotus* on *P. clarkii.* The letters (**A**–**D**) indicated different serial numbers, and the arrows pointed to the momentary behavior of *S. asotus* preying on *P. clarkii*.

**Figure 3 animals-14-01147-f003:**
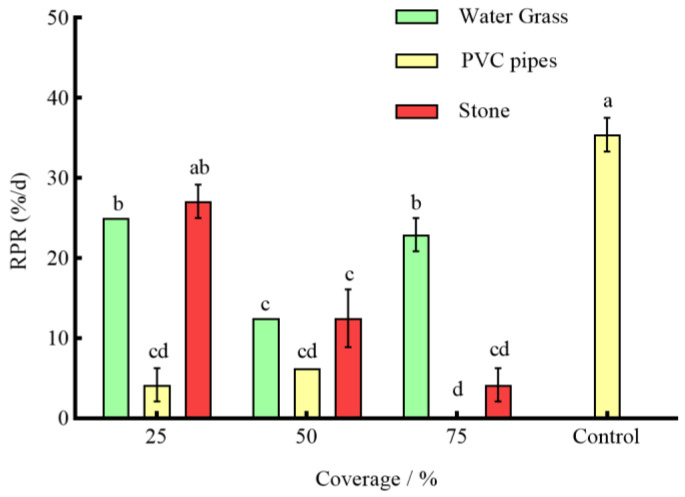
The RPR of *P. clarkii* under different coverage rates. The difference in letters indicated that there were significant differences between groups (*p* < 0.05), and the same applies below.

**Figure 4 animals-14-01147-f004:**
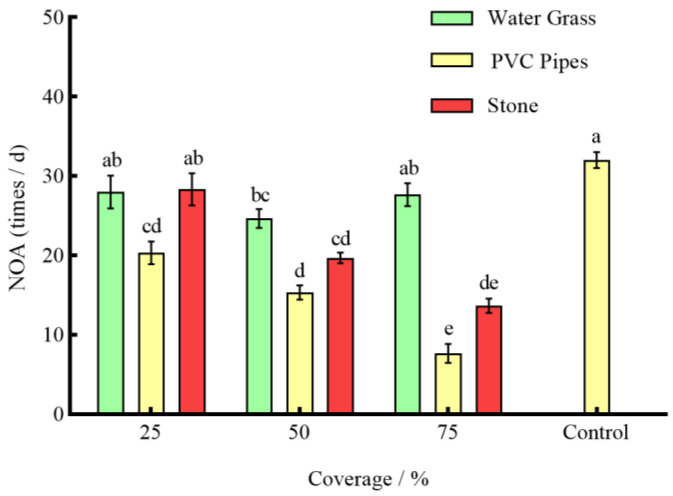
The NOA of *S. asotus* on *P. clarkii* under different coverage rates.

**Figure 5 animals-14-01147-f005:**
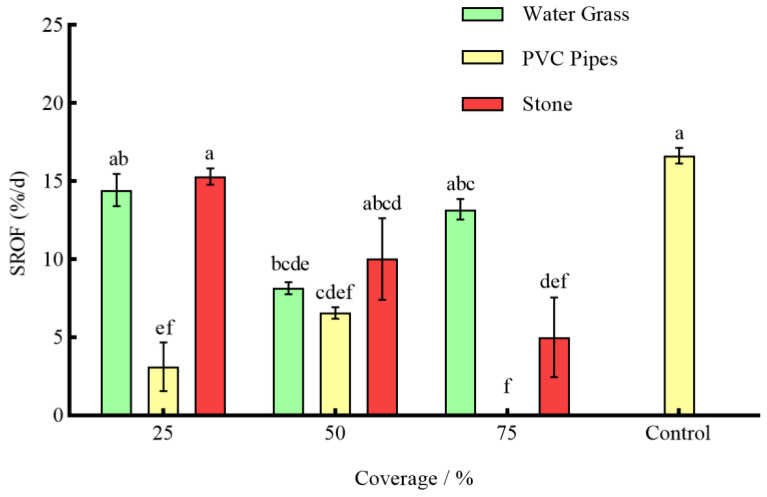
The SROF of *S. asotus* on *P. clarkii* under different coverage rates.

**Figure 6 animals-14-01147-f006:**
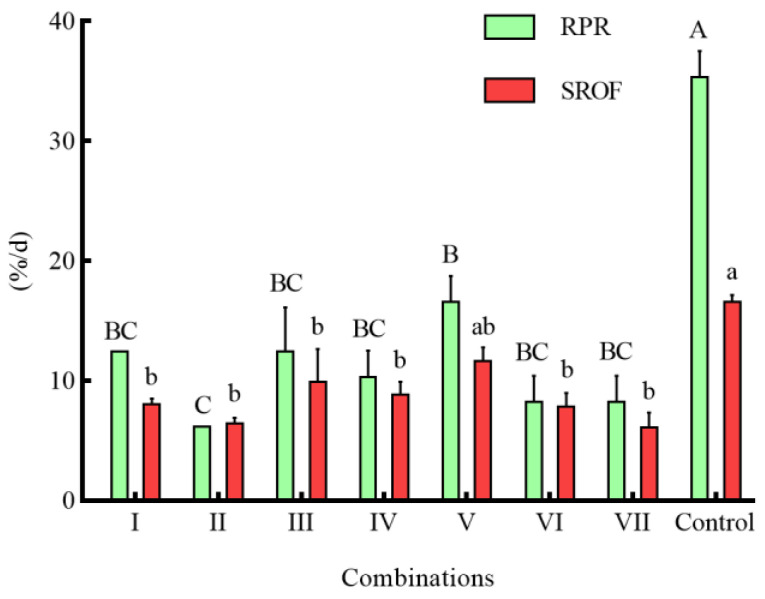
The RPR of *P. clarkii* and SROF of *S. asotus* on *P. clarkii* under different combinations of shelters. The difference in case letters indicated that there were significant differences between groups (*p* < 0.05).

**Figure 7 animals-14-01147-f007:**
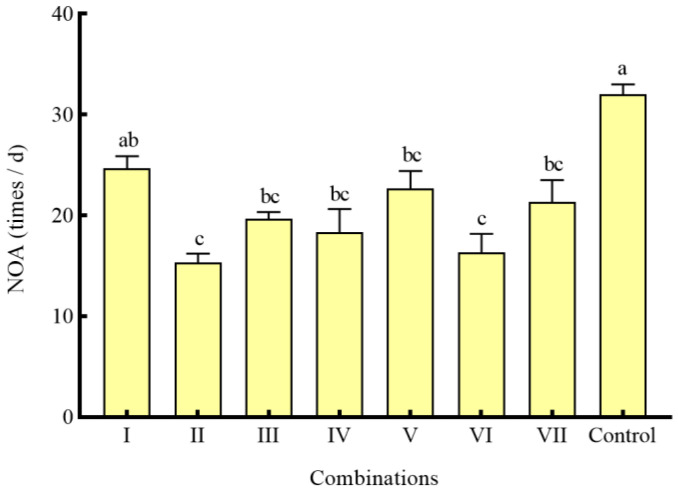
The NOA of *P. clarkii* under different combinations of shelters. The difference in case letters indicated that there were significant differences between groups (*p* < 0.05).

**Table 1 animals-14-01147-t001:** Morphological indices of two experimental aquatic animals.

Species	Number	TL (mm)	BW (g)	MB (mm)	OFL (mm)	BH (mm)	BW (mm)
*S. asotus*	42	311.65 ± 26.53	205.67 ± 30.84	35.03 ± 3.62	27.88 ± 4.11	/	/
*P. clarkii*	672	38.40 ± 1.75	1.94 ± 0.59	/	/	10.48 ± 0.25	9.35 ± 0.77

TL: Total Length; BW: Body Weight; MB: Mouth Breadth; OFL: Fissure Length; BH: Body Height; BW: Body Width.

**Table 2 animals-14-01147-t002:** Combination of three different types of shelters.

Species	Number of *S. asotus*	Number of *P. clarkii*	Number	Shelter Type (Coverage Rate: 50%)		
				Water Grass	PVC Pipes	Stone
*S. asotus*	1	16	I	√		
	1	16	II		√	
	1	16	III			√
	1	16	IV	√	√	
	1	16	V	√		√
	1	16	VI		√	√
	1	16	VII	√	√	√

Note: “√” means that the shelter was set in this experimental group.

**Table 3 animals-14-01147-t003:** Predation rhythm of *S*. *asotus* on *P. clarkii* under different shelter conditions.

Shelter Type		Time Period			
		8~20		20~8	
		RPR (%)	NOA	RPR (%)	NOA
Water grass (WG)	25%	2.08 ± 2.08	7.33 ± 0.66	22.92 ± 1.20 ^a^	20.67 ± 1.69 ^a^
	50%	0	7.33 ± 0.33	12.50 ± 0.00 ^a^	17.33 ± 1.32 ^a^
	75%	4.17 ± 2.05	8.67 ± 0.47	18.75 ± 0.00 ^a^	19.33 ± 1.48 ^a^
PVC pipes (PP)	25%	0	5.67 ± 0.67	4.17 ± 2.05	14.67 ± 0.88 ^a^
	50%	0	5.67 ± 0.33	6.25 ± 0.00 ^a^	9.67 ± 0.67 ^a^
	75%	0	2.33 ± 0.33	0	5.33 ± 0.88
Stone (ST)	25%	4.17 ± 2.05	9.33 ± 0.88	22.92 ± 1.20 ^a^	19.00 ± 1.16 ^a^
	50%	2.08 ± 2.08	7.00 ± 0.58	10.42 ± 1.94 ^a^	12.67 ± 0.33 ^a^
	75%	0	4.00 ± 0.58	4.17 ± 2.05	9.67 ± 0.33 ^a^
Coverage rate: 50%	IV	0	5.67 ± 0.67	10.42 ± 1.94 ^a^	12.67 ± 1.82 ^a^
	V	4.17 ± 2.05	6.67 ± 0.33	12.50 ± 0.00 ^a^	15.33 ± 1.88 ^a^
	VI	2.08 ± 2.08	4.33 ± 0.33	6.25 ± 3.61	12.00 ± 2.00 ^a^
	VII	0	4.66 ± 0.33	8.33 ± 2.05 ^a^	16.67 ± 1.77 ^a^
Control		6.25 ± 0.00	9.67 ± 0.87	29.17 ± 2.06 ^a^	22.33 ± 0.67 ^a^

The difference in letters indicated that there were significant differences between groups (*p* < 0.05), and the same applies below.

## Data Availability

The data presented in this study are available in the article. Further information is available upon request from the corresponding author.

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
