# Peer review of "The Influence of Shelter Type and Coverage on Crayfish (Procambarus clarkii) Predation by Catfish (Silurus asotus): A Controlled Environment Study"

_animals, 2024, doi:10.3390/ani14081147_

Round 1

Reviewer 1 Report

Comments and Suggestions for Authors

The MS is generally good, but there are still some areas that need to be revised and improved.

1. The materials and methods section should be supplemented with statistical analysis.

2. What are the sizes and specifications of the PVC pipes and stones used in the experiment? Does the size and shape of the shelter affect the predation relationship?

3. Are all the experimental results in Figure 3- Figure 7 dark conditions or light conditions?

4. The result of this experiment is that PVC pipe has the best sheltering effect on P. clarkii. What is the significance of this to the germplasm of  P. clarkii in Poyang Lake?

5. The author abbreviation of reference 16 needs to be corrected.

Comments on the Quality of English Language

Further revisions are needed for English writing.

Reviewer 2 Report

Comments and Suggestions for Authors

Line33-35It is not certain that there is a causal relationship between closure of fishing and P. clarkii in Poyang Lake exhibited a trend towards smaller individuals. This statement should be revised or corresponding references should be provided.

Line114Why is the behavior observation of only 50% coverage combination?

Line115The length of area An and B is not clear

Line107Lack of definition of attack behavior

Line 109: Lack of description of statistical methods

Line173 and Line197Lack of standard error in fig.3 and 6.

Reviewer 3 Report

Comments and Suggestions for Authors

This manuscript presents the results of an investigation into the effect of shelters on the predation of crayfish (Procambarus clarkii) by the catfish species Silurus asotus. The crayfish were used as prey, and the catfish were the predators, with shelters made of PVC pipes, water grass, and stone being tested.

While the study provides valuable insights, its focus solely on shelter type and coverage may limit the generalizability of the results. Other factors likely influence predation success, such as the body size disparity between crayfish and catfish. Larger catfish might overcome some shelter limitations, while smaller prey might utilize even narrow crevices effectively. Additionally, water clarity, presence of alternative prey, and time of day could influence predator-prey interactions.

Furthermore, the study assumes that catfish readily target crayfish as prey. However, it's crucial to consider prey preference in natural settings. Research on habitat overlap and foraging behavior of Silurus asotus is needed to determine if crayfish are their preferred prey. Perhaps other fish species or invertebrates might be a more typical food source for catfish, impacting their motivation to pursue crayfish, especially those with effective shelters. The study design might not capture the full range of hunting strategies catfish might employ in a real riverbed compared to a controlled tank environment.

Despite the limitations discussed above, this work presents interesting findings and holds potential for further exploration. Here are some suggestions for strengthening the manuscript.

-The current title “The effect of Silurus asotus predation on Procambarus clarkii under different shade environments” should be revised, perhaps this would be more appropriate: "The Influence of Shelter Type and Coverage on Crayfish Predation by Catfish: A Controlled Environment Study,"

-The discussion section can be significantly enhanced by elaborating on the identified limitations. Consider exploring how factors like body size disparity, water clarity, presence of alternative prey, and time of day might influence predation rates in a natural setting. Additionally, addressing the lack of information on prey preference and the need for research on habitat overlap and Silurus asotus' foraging behavior would further strengthen the discussion. It's also crucial to emphasize the potential discrepancies between catfish hunting strategies in a controlled environment compared to a natural riverbed.

-The conclusion can be improved by summarizing the key findings on shelter type and coverage's impact on predation rates. However, it's equally important to acknowledge the limitations discussed earlier and emphasize the need for further research incorporating these factors. Briefly summarize the key findings of the study regarding the impact of shelter type and coverage on predation rates. Clearly state the limitations identified in the discussion, emphasizing the need for further research that incorporates these factors. End with a more cautious statement about the applicability of the findings to natural settings.

 The abstract should be revised to include the changes suggested above.

Comments on the Quality of English Language

there are few spelling mistakes (see line 10, the word abstract for example). The title should include the common names of the two species ( crayfish and catfish).

Round 2

Reviewer 3 Report

Comments and Suggestions for Authors

thank you for your effort to impove your manuscipt.